# Iron–Gold Nanoflowers: A Promising Tool for Multimodal Imaging and Hyperthermia Therapy

**DOI:** 10.3390/pharmaceutics14030636

**Published:** 2022-03-14

**Authors:** Evangelia Christou, John R. Pearson, Ana M. Beltrán, Yilian Fernández-Afonso, Lucía Gutiérrez, Jesús M. de la Fuente, Francisco Gámez, María L. García-Martín, Carlos Caro

**Affiliations:** 1BIONAND—Centro Andaluz de Nanomedicina y Biotecnología (Junta de Andalucía-Universidad de Málaga), C/Severo Ochoa, 35, 29590 Málaga, Spain; lillo0ochristou@gmail.com (E.C.); jrpearson@bionand.es (J.R.P.); 2Departamento de Ingeniería y Ciencia de los Materiales y del Transporte, Escuela Politécnica Superior, Universidad de Sevilla, Virgen de Á frica 7, 41011 Sevilla, Spain; abeltran3@us.es; 3Instituto de Nanociencia y Materiales de Aragón (INMA), CSIC-Universidad de Zaragoza, 50009 Zaragoza, Spain; yfdezafonso@gmail.com (Y.F.-A.); lu@unizar.es (L.G.); jmfuente@unizar.es (J.M.d.l.F.); 4Biomedical Research Networking Center in Bioengineering, Biomaterials & Nanomedicine (CIBER-BBN), 28029 Madrid, Spain; 5Departamento de Química Física, Facultad de Ciencias Químicas, Universidad Complutense de Madrid, 28040 Madrid, Spain; frgamez@ucm.es

**Keywords:** gold–iron nanoparticles, nanoflowers, MRI, CT, multimodal imaging, photothermal therapy, magnetic hyperthermia

## Abstract

The development of nanoplatforms prepared to perform both multimodal imaging and combined therapies in a single entity is a fast-growing field. These systems are able to improve diagnostic accuracy and therapy success. Multicomponent Nanoparticles (MCNPs), composed of iron oxide and gold, offer new opportunities for Magnetic Resonance Imaging (MRI) and Computed Tomography (CT) diagnosis, as well as combined therapies based on Magnetic Hyperthermia (MH) and Photothermal Therapy (PT). In this work, we describe a new seed-assisted method for the synthesis of Au@Fe Nanoparticles (NPs) with a flower-like structure. For biomedical purposes, Au@Fe NPs were functionalized with a PEGylated ligand, leading to high colloidal stability. Moreover, the as-obtained Au@Fe-PEG NPs exhibited excellent features as both MRI and CT Contrast Agents (CAs), with high r_2_ relaxivity (60.5 mM^−1^⋅s^−1^) and X-ray attenuation properties (8.8 HU mM^−1^⋅HU). In addition, these nanoflowers presented considerable energy-to-heat conversion under both Alternating Magnetic Fields (AMFs) (∆T ≈ 2.5 °C) and Near-Infrared (NIR) light (∆T ≈ 17 °C). Finally, Au@Fe-PEG NPs exhibited very low cytotoxicity, confirming their potential for theranostics applications.

## 1. Introduction

Advances in the management of challenging diseases, such as cancer, require new approaches to improve the specificity and sensitivity of early diagnosis and provide more efficient therapies. Medical imaging is a cornerstone of clinical diagnosis and is increasingly used to guide surgical and radiation-based interventions. Particularly, Magnetic Resonance Imaging (MRI) and Computed Tomography (CT) are extensively used because of their excellent anatomical resolution [1,2]. Both techniques often require the administration of external Contrast Agents (CAs) for improved sensitivity and/or selectivity [3,4]. However, side effects, including induced toxicity and allergic reactions, have been reported for some CAs currently in clinical use [5,6,7]. As for cancer treatment, systemic drug administration is a “gold standard” therapeutic approach [8], but its side effects, due to its lack of specificity, remain a major source of concern [9]. Hyperthermia Treatment (HT), which consists of a local increase in tissue temperature up to 40–44 °C, has been demonstrated to boost the effectiveness of systemic therapy with promising results [10,11]. HT techniques include radiofrequency, microwave ablation, high-intensity focused ultrasound, and laser ablation [12]. However, reported HT effectiveness has varied, and there is a lack of both adequate treatment experience and comprehensive statistical studies to optimize its clinical application [13,14]. Thus, further research is required to enhance HT effectiveness and improve HT targeting against malignant tissues to reduce non-specific side effects.

Nanotechnology can be defined as the exploitation of the distinctive properties exhibited by matter when organized at the nanometric level [15]. Its application to medicine, nanomedicine, has the potential to drive the development of new devices, CAs, and drugs with improved specificity, efficiency, accuracy, and personalization [16,17]. A wide range of NPs has been developed for medical applications. For example, Superparamagnetic Iron Oxide Nanoparticles (SPIONs) have demonstrated exceptional characteristics as MRI CAs [18,19,20,21] as well as an effective energy-to-heat converter in the presence of an Alternating Magnetic Field (AMF), making them suitable for HT [22,23,24,25]. In fact, SPIONs are being evaluated in clinical trials, ongoing or finished, for a broad spectrum of pathologies [26,27]. Another widely studied example is Gold Nanoparticles (AuNPs) [28]. AuNPs exhibit good X-ray attenuation, which makes them suitable as CT CAs [29]. In addition, due to their tunable Surface Plasmon Resonance (SPR), AuNPs can possess strong optical absorption in the Near-Infrared (NIR) range (800–1200 nm). The NIR range is of particular interest as it corresponds to wavelengths for which tissues are moderately transparent, often referred to as the biological window [30,31]. This attribute can also be exploited to generate localized heat by a process termed light-to-heat transduction [32].

Recently, Multicomponent Nanoparticles (MCNPs) have attracted substantial attention because of their ability to combine the properties of different inorganic materials in a single nanoscale entity [33,34,35,36,37]. In fact, improved diagnostic accuracy has been demonstrated using MCNP-mediated multimodal imaging [38,39], as well as increased patient safety by reducing the amount of CAs injected. Combined therapies are an area attracting considerable attention for the treatment of numerous pathologies [40,41,42]. It is worth noting that, since MCNPs frequently present different composition surfaces, it is possible to introduce multiple functionalizations, each with specific active molecules and distinct biological activities [43].

Here, we report a new seed-assisted synthetic route for producing Au@Fe NPs. The effect of the iron precursor concentration and its impact on MCNP morphology is also addressed and discussed. Further, the most homogeneous Au@Fe NPs were functionalized with a PEGylated ligand to promote their stabilization in aqueous media. Then, PEGylated Au@Fe NPs were thoroughly characterized to determine their magnetic properties, magnetic relaxivity, X-ray attenuation efficiency, and heat conversion capabilities when exposed to either NIR irradiation or AMFs. Finally, the cytotoxicity of PEGylated Au@Fe NPs was evaluated in cell culture to determine their suitability for biomedical applications. To the best of our knowledge, this is the first time that both multimodal diagnostic and multimodal therapeutic capabilities have been demonstrated for Au@Fe NPs with HD below 100 nm and with no detectable cytotoxicity at high concentrations.

## 2. Materials and Methods

### 2.1. Materials

Gold (III) acetate, iron (III) chloride, sodium oleate, oleic acid 99%, oleylamine, 1,2-hexadecanediol, gallic acid, polyethylene glycol 3000 Da, 1-octadecene, triethylamine, 4-dimethylaminopyridine, Dicyclohexyl Carbodiimide (DCC), 3-[4,5-dimethylthiazol-2yl]-2,5-diphenyl tetrazolium bromide (MTT), Phosphate-Buffered Saline (PBS), and ethanol absolute were obtained from Sigma-Aldrich (St. Louis, MO, USA). Hoechst 33342 and Propidium Iodide (PI) were purchased from Merck. Dimethyl Sulfoxide (DMSO), toluene, acetone, hexane, chloroform, dichloromethane, and tetrahydrofuran were supplied by Acros organics. Dulbecco’s Modified Eagle Medium (DMEM), Fetal Bovine Serum (FBS), L-glutamine, and penicillin/streptomycin solution were obtained from Gibco. All reagents were used as received without further purification. Milli-Q water (18.2 MΩ, filtered with filter pore size 0.22 µM) was obtained from Millipore.

### 2.2. Synthesis of Iron–Gold Nanoflowers

#### 2.2.1. Synthesis of Iron Oleate

The iron oleate precursor was synthesized following a previously described synthetic route [44].

#### 2.2.2. Synthesis of Au Seeds

AuNP seeds were synthesized as previously described by Tancredi et al. [45]. Briefly, 50 mg of gold (III) acetate and 100 mg of 1,2-hexadecanodiol were dissolved in a mixture of 0.8 mL oleic acid, 0.6 mL oleylamine, and 5 mL 1-octadene. The solution was heated under vacuum up to 120 °C (heating rate of 5 °C⋅min^−1^) for 30 min. AuNPs were then cooled and washed 3 times with a 1:1 ratio of ethanol/acetone to remove any unreacted reagents.

#### 2.2.3. Synthesis of Au@Fe NPs

One milliliter of the Au seeds (≈10 mg of metallic Au, 0.05 mmol) was combined with 0.63 mL (2 mmol) oleylamine, 0.66 mL (2 mmol) oleic acid, 0.645 g 1,2-hexadecanediol (2.5 mmol), 10 mL of 1-octadecene, and the proper amount of iron oleate. The mixture was magnetically stirred under a flow of nitrogen and then heated to 120 °C for 20 min. The temperature was then increased to 200 °C and maintained for 120 min. In a third heating step, the temperature was increased again to 315 °C (heating rate of 5 °C⋅min^−1^) and maintained for 30 min. Finally, the suspension was cooled down, washed 3 times with a 1:1 ethanol/acetone mixture, and dispersed in 10 mL of toluene.

### 2.3. Functionalization of Au@Fe NPs

The synthetic route for the PEGylated ligand and subsequent ligand exchange were conducted as described previously [18,46]. Detailed protocols are provided in the Appendix A.

### 2.4. Characterization

Au@Fe NPs were characterized by several techniques, namely, Transmission Electron Microscopy (TEM), Scanning Transmission Electron Microscopy (STEM), UV–Vis spectroscopy, X-ray Diffraction (XRD), Inductively Coupled Plasma High-Resolution Mass Spectroscopy (ICP-HRMS), Fourier Transform Infrared Spectroscopy (FTIR), Dynamic Light Scattering (DLS), magnetometry, measurement of optical Specific Loss Power (oSLP) and magnetic SLP (mSLP), photothermal conversion, relaxivities at different magnetic fields, and X-ray attenuation efficiency. Complete protocols are described in detail in the Appendix A.

### 2.5. Cytotoxicity Assays

#### 2.5.1. Cell Culture

As models, HFF-1 human fibroblast cells were selected. Cells were cultured at 37 °C in an incubator with a humidified atmosphere containing 5% CO_2_. DMEM supplemented with 2 mM of L-glutamine, 10% Fetal Bovine Serum (FBS), and 1% penicillin/streptomycin was used as growth medium.

#### 2.5.2. Cytotoxicity Assays

Cytotoxicity was evaluated in HFF-1 cells by LIVE–DEAD and MTT assays. Several markers associated with cellular viability, including mitochondrial activity, cell morphology, and plasma membrane integrity, were evaluated. Protocols details are provided in the Appendix A.

### 2.6. Statistical Analysis

The statistical analysis was performed using the SPSS package (SPSS Inc., Chicago, IL, USA). Cell viability values are shown as mean ± standard deviation (SD). Student’s *t*-tests or one-way analysis of variance were used to determine significant differences. The level of significance was set at *p* < 0.05.

## 3. Results and Discussion

### 3.1. Synthesis and Characterization of Au@FeNPs

Inorganic NP synthesis is a complex process that is not yet fully understood. LaMer growth followed by Ostwald ripening is the most frequently accepted process of NP nucleation and growth [47]. For the synthesis of these novel Au@Fe NPs, we used a seed-mediated methodology, whereby iron atoms should react at the surface of gold nuclei, leading to NP growth. Gold seed growth resulted in spherical ≈ 12 nm NPs (Appendix A). We performed an initial growth process using 0.5 mmol of iron precursor, which resulted in the formation of new iron nuclei (Appendix A). Thus, this iron precursor concentration reaches a critical limiting supersaturation level. By decreasing the amount of iron to 0.25 mmol, homogeneous nanoflower-like structures were obtained (Figure 1a and Appendix A). Moreover, STEM analysis revealed that the sepal (core) was composed of a chemical element with a higher Z number than that found in nanoflower petals (Figure 1a). HAADF and EDX measurements in STEM mode confirmed that the sepal was fully composed of gold, whereas only iron atoms were found in the petals (Figure 1b). Nanoflower size distribution was assessed by Transmission Electron Microscopy (TEM), where we observed an average size of 11.7 ± 2.7 nm and 33.0 ± 6.1 nm, for the core and whole nanoflower, respectively (Figure 1c). Therefore, Au core size was almost the same as a single Fe petal. Powder XRD was used to characterize the crystalline structure of Au@Fe NPs, revealing intense Au peaks indexed to (111), (200), and (220) phases [48]. In addition, intense Fe peaks were indexed to (220), (311), (400), (511), and (440) phases, which can be attributed to either γ–Fe_2_O_3_ or Fe_3_O_4_ [49]. Therefore, XRD analysis confirmed the crystalline structure of the Au@Fe NPs (Figure 1d).

Regarding the optical properties of Au@Fe NPs, moderate extinction in the 500–1000 nm range was detected, with a maximum SPR value centered at 535 nm (Figure 2a). Although gold nanospheres usually present a narrow SPR in the visible region [50], the broad extinction spectrum exhibited by Au@Fe NPs can be explained by the interaction of SPR with the iron petals. Importantly, this NIR absorption covers the biological window, which potentially enables the use of Au@Fe NPs for biological studies using optical techniques. However, these Au@Fe NPs were unstable in aqueous media, making them unsuitable for biomedical applications. To address this issue, we performed a ligand exchange process with a PEGylated derivative. We found no evidence that this process affected the size or shape of the resulting Au@Fe-PEG NPs (Appendix A), only that the UV–Vis spectrum shifted slightly to a lower wavelength (Appendix A). Strong interaction between the PEGylated derivative and the NP surface was confirmed by FTIR spectroscopy, where the main PEG peaks were assigned to the Au@Fe-PEG NPs spectrum (Figure 2b), as previously described by us [17]. In addition, the Hydrodynamic Diameter (HD) and zeta potential measured in saline solution were 58.1 ± 8.4 nm (Figure 2c) and −13.7 ± 0.7 mV, respectively. This HD is in the 10–200 nm range, often described as suitable for biomedical applications, especially for cancer theranostics [51]. Moreover, their zeta potential is expected to lead to high stability, according to DLVO theory [52]. Such stability was confirmed by measuring the HD values over a month without any evidence of flocculation or aggregation (Figure 2d).

The magnetic hysteresis loop of Au@Fe-PEG NPs exhibited superparamagnetic behavior, with negligible Coercivity Field (Hc) (Figure 3a). Next, we evaluated the features of Au@Fe-PEG NPs to see if they would be suitable as CAs for multimodal imaging. On the one hand, their magnetic transverse relaxivity, r_2_, was determined to be 47.8 mM^−1^⋅s^−1^ and 60.5 mM^−1^⋅s^−1^ at 1.44 and 9.4 T, respectively (Figure 3b). It is worth noting that these calculated r_2_ values were in the range of similar MCNPs [53], and can be explained by the quantum mechanical outer-sphere theory [54]. The longitudinal relaxivity, r_1_, was also calculated at 1.44 T, corresponding to 1.8 mM^−1^⋅s^−1^ (Appendix A), which results in an r_2_/r_1_ ratio of 26.5, implying that Au@Fe-PEG NPs behave as a T_2_ CA [55]. On the other hand, the X-ray attenuation coefficient was determined by plotting Hounsfield Unit (HU) variation as a function of Au concentration. The slope of Au@Fe-PEG NPs (8.8 HU mM^−1^⋅HU [Au]) was higher than that of iohexol (2.69 HU mM^−1^⋅HU [I]), which is a commonly used clinical CA [7] (Figure 3c). Therefore, our data strongly support the potential of Au@Fe-PEG NPs as a multimodal CA for MRI and CT. Finally, local heat generation triggered by NIR light and alternating magnetic fields was assessed. Both induced HT processes resulted in a temperature increase when Au@Fe-PEG NPs were exposed to AMFs or light, recovering slowly to basal temperature when either the laser or the AMF were switched off (Figure 3d). This behavior was reproducible at least for the four measured cycles. However, the obtained temperature increase in both approaches was significantly different after 10 min of exposure (∆T was ≈ 17 °C in the case of optical HT, whereas a lower ∆T ≈ 2.5 °C was recorded for magnetic HT). The experimental temperature rise was fitted to the analytical solution of the Fourier equation of a point-like source (see red dashed line in Figure 3d) [56]:(1)$$\Delta T\left(t\right)=\Delta T_{\infty }\left[1+\left(\frac{t}{2\tau }-1\right)erf\left(\sqrt{\frac{t}{\tau }}\right)-\sqrt{\frac{t}{\pi \tau }}exp\left(\frac{t}{\tau }\right)\right]$$
where τ is the relaxation time and erf(x) is the error function of argument x. The temperature increase at the steady-state was estimated to be ∆T_∞_ ≈ 26.3 °C from the fitting of the experimental data to the Fourier equation of a point-like source (see red dashed line in Figure 3d). The oSLP, calculated from the temperature increase rate at *t* = 10 min, was 3.12 kW∙g^−1^, and the photothermal conversion was ≈36% according to previously reported methodology [57], whereas the obtained mSLP was 205.7 W⋅g^−1^. This mSLP was 20% higher compared to similar Au-Fe hetero-nanostructures [58]. These results are in good agreement with the linear response theory model, which predicts that 22 nm spherical NPs should provide the optimal mSLP response [22]. The significantly higher than expected mSLP values presented by Au@Fe-PEG NPs might be related to a higher effective anisotropy, which, as previously reported, is a crucial parameter for the magnetic HT performance of magnetic nanoparticles [59,60]. Further, mSLP was also tested at lower magnetic field amplitude (304 Gauss), resulting in ≈90 W/g, which is still higher than the one exhibited by other magnetic NPs at comparable magnetic field amplitude and higher frequencies [61]. Moreover, a frequency of 182 kHz and a magnetic field amplitude of 280 Gauss have been recently reported for in vivo experiments [62]. Additionally, Au@Fe-PEG NPs displayed excellent thermal stability under either NIR irradiation or AMF treatment, maintaining similar maximum temperatures over four cycles (5 min on followed by 30 min off). Thus, these results confirmed local heat generation by both optical and magnetic HT due to the intrinsic properties of Au@Fe-PEG NPs.

### 3.2. In Vitro Cytotoxicity

Besides the promising characteristics described above for Au@Fe-PEG NPs as contrast/therapeutic agents, a comprehensive evaluation of their cytotoxicity is crucial to elucidate their true potential for biomedical applications. Thus, LIVE–DEAD and MTT assays were conducted in cultures of HFF-1 fibroblasts. We observed no statistically significant changes (*p* < 0.05) in the total number of cells, dead cell percentage, or mitochondrial activity compared to control cells even at high concentrations (up to 50 µg/mL Fe + Au). Higher concentrations, up to 100 µg/mL Fe + Au, were also tested, showing no effect on mitochondrial activity or cell death (Appendix A). However, this high concentration of Au@Fe-PEG NPs interfered with the cell counting of the high content imaging system used for these analyses (Perkin Elmer Operetta) and therefore was not included in the results in Figure 4. These cytotoxicity results confirm that Au@Fe-PEG NPs exhibit no significant cytotoxicity under normal conditions, further supporting their potential as theranostic nanoplatforms.

## 4. Conclusions

In this work, gold@iron oxide NPs with a nanoflower-like structure were synthesized by a seed-mediated growth methodology. The composition and spatial distribution of the chemical atoms were confirmed by HAADF-EDX in STEM mode and XRD. After a ligand exchange process with a PEGylated derivative, the nanoflowers exhibited high colloidal stability in saline for 30+ days. In addition, Au@Fe-PEG NPs presented high r_2_ relaxivity and X-ray attenuation properties, as well as a substantial level of heat conversion in response to NIR photons or AMFs. Furthermore, Au@Fe-PEG NPs revealed very low cytotoxicity. Altogether, our data indicate that Au@Fe-PEG NPs are promising candidates for in vivo imaging-guided therapeutic applications, which are currently under investigation. Further experiments will be devoted to the functionalization of these nanoflowers with specific ligands targeted against tumor cell markers to enable in vivo molecular imaging and improve their potential for combined hyperthermia treatment.

## Figures and Tables

**Figure 1 pharmaceutics-14-00636-f001:**
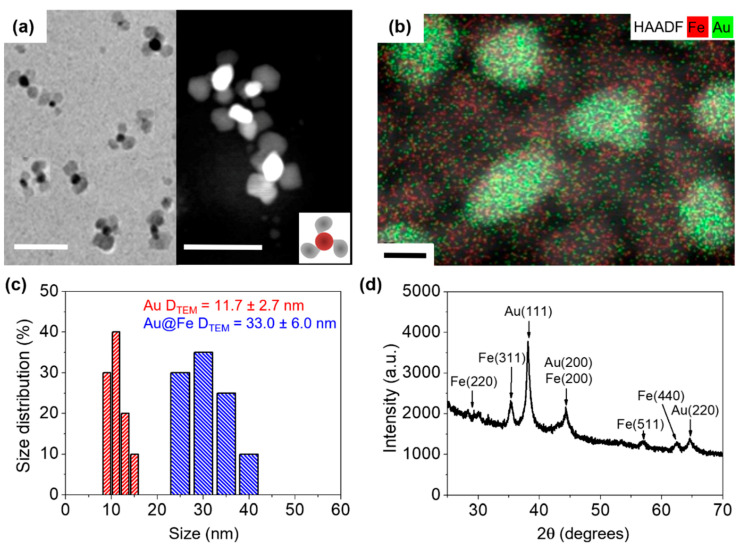
(**a**) Representative TEM (left) and STEM (right) images of Au@Fe NPs. The inset represents a scheme of the obtained NPs. Scale bar corresponds to 50 nm. (**b**) Spatial distribution of metal atoms in the Au@Fe NPs as measured by HAADF and EDX. Scale bar corresponds to 10 nm. (**c**) Size distribution derived from counting at least 100 NPs in TEM images of Au seeds (red) and Au@Fe NPs (gray). (**d**) X-ray powder diffraction (XRD) pattern of Au@Fe NPs.

**Figure 2 pharmaceutics-14-00636-f002:**
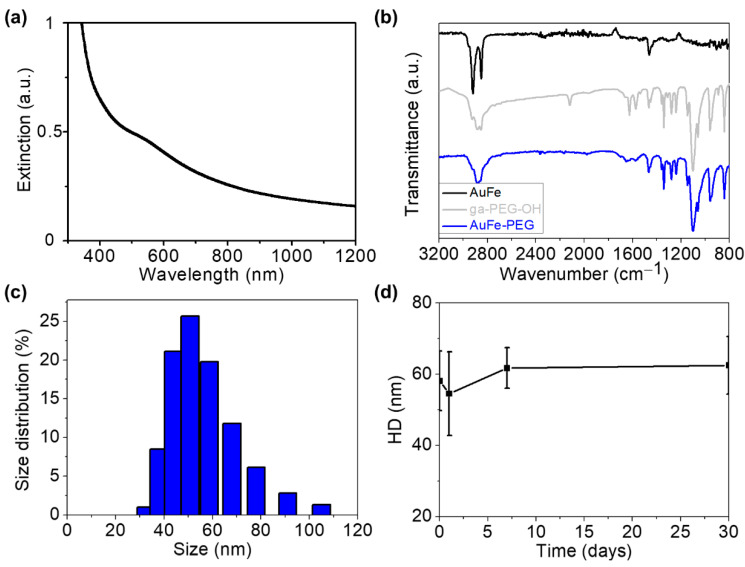
(**a**) UV–Vis–NIR spectrum of Au@Fe NPs. (**b**) FTIR spectra of oleic acid capped Fe@Au NPs (black), PEGylated ligand (gray), and PEGylated Fe@Au NPs (blue). (**c**) Hydrodynamic diameters in saline of Au@Fe-PEG NPs. (**d**) Stability measured over time.

**Figure 3 pharmaceutics-14-00636-f003:**
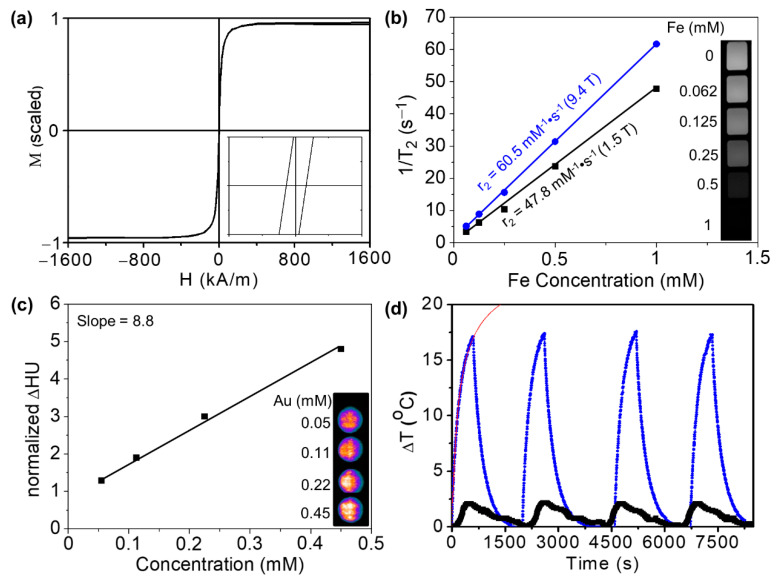
(**a**) Hysteresis loop of Au@Fe-PEG NPs measured at 300 K scaled to the maximum magnetization value. (**b**) Plot of relaxation rate (1/T_2_) vs. the concentration of Fe in the Au@Fe-PEG NP suspension determined at 1.44 T (black dots) and 9.4 T (blue dots), and the linear fits (black and blue lines) whose slopes correspond to the transverse relaxivities (r_2_) of Au@Fe-PEG NPs at both magnetic fields. (**c**) X-ray attenuation measured as the linear relationship between the normalized ∆HU of the CT images and the Au concentration of the Au@Fe-PEG NP solutions. (**d**) Thermal stability test of Au@Fe-PEG NPs measured under NIR irradiation (blue) and AMF (black). The dashed red line is a fit to the solution of the Fourier transform of a point-like source.

**Figure 4 pharmaceutics-14-00636-f004:**
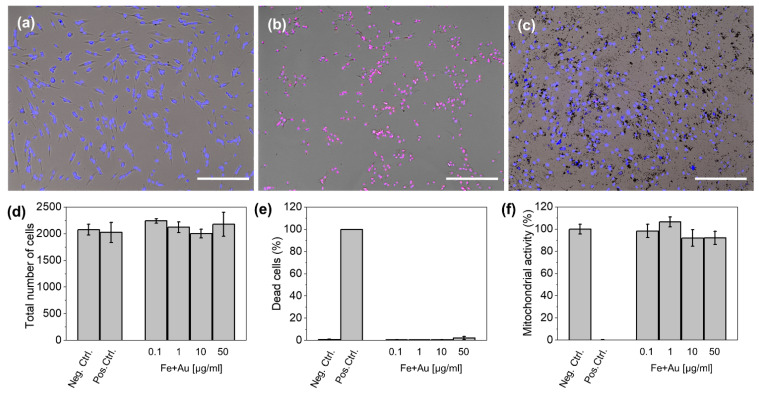
Representative optical microscopy images of HFF-1 cells incubated with Au@Fe-PEG NPs: (**a**) negative control, (**b**) positive control, (**c**) cells incubated with 50 µg/mL Fe + Au. Images show a merge of brightfield (gray), Hoechst 33342 (blue), and PI (red). Scale bars correspond to 100 µm. Quantification of total number of cells per well, (**d**) percentage of dead cells, (**e**) mitochondrial activity (**f**) of cells exposed to increasing concentration of Au@Fe-PEG NPs.

## Data Availability

The data presented in this study are available on request from the corresponding author.

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
