# Peer review of "Iron–Gold Nanoflowers: A Promising Tool for Multimodal Imaging and Hyperthermia Therapy"

_pharmaceutics, 2022, doi:10.3390/pharmaceutics14030636_

Round 1

Reviewer 1 Report

The paper is on hybrid Au@Fe nanoflowers. The authors synthesized these nanoparticles and coated them with PEG molecules to apply in imaging and hyperthermia applications.  The paper is well-organized and scientific. Although, there are some points that must be considered to improve it to fit the journal:

  1. The novelty of work must be mentioned in the introduction (what makes this work special)
  2. Nano flower must include in keywords
  3. Line 39, P1: this sentence is not relevant to the previous one! please rewrite this part
  4. 39℃ is not considered as hyperthermic temperature. please check this.
  5. “Indeed, HT has 44 shown per se promising results, with enhanced patient survival observed…” is vague. Please rewrite it.
  6. the authors have not specified which hyperthermia modality they are talking about! if they mean magnetic hyperthermia: there is no treatment experience on this yet but, If they mean RF hyperthermia: there are many studies on this method. Please specify the hyperthermia and change this sentence. Line 46, page 1
  7. line 64, page 2: recently must be started from next paragraph
  8. page 3, line 123: change the headline to characterization…
  9. page 5, line 181: with a maximum value centered at 535 nm.. does not match with Figure 2a
  10. line 226: from the fitting of the experimental data 226 to the Fourier equation of a point-like source (see red dashed line in Figure 3d). please explain this sentence.
  11. for magnetic hyperthermia, SLP must be used not SAR. please change SAR to SLP
  12. 50 ug/ml Fe+Au seems to not have a toxicity effect regarding PEG coating. higher concentrations such as 100-500 ug/ml must be tested.
  13. the authors have not used high concentrations of Fe_Au nanoparticles. they cannot conclude this: Furthermore, Au@Fe-PEG NPs revealed very low cytotoxicity. Line 277
  14. the conclusion is too short; explain future studies using these particles or limitations of your study
  15. Tissue was not used in this study. Why did the authors explain the ICP-MS for the tissues in the supplementary file?
  16. magnetic field amplitude must be mentioned as kA/m

Reviewer 2 Report

The article is interesting and deserves publication. I have few following comments: 1. Authors worked with 12 nm seed Au. Did you get different size of seed? Is it work with another size? Could you present uv-vis for seed solution for comparing? 2. The field strengths used for MHT is high 580 Gauss. For in vivo/in-vitro applications, better to use magnetic field of ~150 Gauss. Do the particles heat at lower field parameters? 3. From the paper it is not clear how calculated o-SAR and m-SAR: per 1g of Iron, Iron oxide, Iron oxide/Au or particles? How much organic part in the samples? TGA or another analysis should be done for this. 4. Could you provide with magnetisation value for your samples? Correlate calculation with mass per oxide or metal (and with SAR calculation as well). 5. Please refer to articles on IONP that have shown excellent MHT heating efficacy and MRI T1 contrast agent: - Storozhuk, L., Besenhard M. O., Mourdikoudis, S., LaGrow, A. P., Lees, M.R., Tung, L. D., Gavriilidis, A., Thanh, N. T. K* (2021) Simple and Fast Polyol Synthesis of Stable Iron Oxide Nanoflowers with Exceptional Heating Efficiency. Journal of Applied Materials and Interface. 13: 45870−45880. - Besenhard M. O., Panariello, L., Kiefer, C., LaGrow, A. P., Storozhuk, L., Perton F., Begin, S., Damien Mertz, D., Thanh, N. T. K.* and Gavriilidis, A. (2021) Small Iron Oxide Nanoparticles as MRI T1 Contrast Agent: Scalable Inexpensive Water-Based Synthesis Using a Flow Reactor. Nanoscale. 13: 8795-8805.

Reviewer 3 Report

The principal objective of this study was to evaluate the Iron-Gold nanoflowers as theranostic agents. It is an interesting study with valuable results in in vitro; however, there are some major concerns;

  1. The manuscript needs to be checked again for English grammar.
  2. A graphical abstract could be helpful for better illustration of study procedure.
  3. The Materials and Methods section needs more details as well as study procedure and protocols, the applied materials, apparatus and kits with properties and Cat. No. in both NPs synthesis/characterization and hyperthermia assessment, and in vitro toxicity tests.
  4. It’s a good idea to evaluate the effects of these NPs on ROS generation and DNA strands.
  5. The NPs concentration including 0.1, 1, 10 and 50 μg/ml is not sufficient. It needs more different concentrations. Below papers are good Ref. for NPs concentrations;

- Combinatorial effects of radiofrequency hyperthermia and radiotherapy in the presence of magneto‐plasmonic nanoparticles on MCF‐7 breast cancer cells

- Selective heat generation in cancer cells using a combination of 808 nm laser irradiation and the folate-conjugated Fe2O3@Au nanocomplex

  1. Which concentration of NPs reported as optimum dose for evaluation of theranostics effects and cytotoxic assays?
  2. It needs to improve the resolution of Figures.
  3. It’s a good idea you treat the cells with optimum dose and then evaluate theranostics effects.
  4. In the Introduction, the authors need to elaborate on the role of nanoparticles particularly  the Iron oxide and gold nanoparticles as theranostic tools in the fight against different diseases by citing and briefly discussing the following papers (DOI: 1016/j.molliq.2018.05.105, DOI: 10.3390/nano11082033, DOI: 10.1016/j.molstruc.2018.07.092, DOI: 10.1016/j.molstruc.2018.04.016).
  5.  Pay attention on the more interpretation of the experimental results. Only their presentation is not enough for a scientific paper.
  6. The abstract and conclusion parts must be more informative by including more mathematical findings.
  7. Please more focus on providing information regarding the clinical trials of magnetic induction hyperthermia for cancer treatment in the Introduction

Round 2

Reviewer 1 Report

The paper is acceptable in the revised form. all comments are properly addressed.

Reviewer 3 Report

It is acceptable now.